# Safe LoRA: the Silver Lining of Reducing Safety Risks when Fine-tuning Large Language Models

**Chia-Yi Hsu**
National Yang Ming Chiao Tung University
Hsinchu, Taiwan
chiayihsu8315@gmail.com

**Yu-Lin Tsai**
National Yang Ming Chiao Tung University
Hsinchu, Taiwan
uriah1001@gmail.com

**Chih-Hsun Lin**
National Yang Ming Chiao Tung University
Hsinchu, Taiwan
pkevawin334@gmail.com

**Pin-Yu Chen**
IBM Research
New York, USA
pychen@ibm.com

**Chia-Mu Yu**
National Yang Ming Chiao Tung University
Hsinchu, Taiwan
chiamuyu@gmail.com

**Chun-Ying Huang**
National Yang Ming Chiao Tung University
Hsinchu, Taiwan
chiamuyu@gmail.com

## Abstract

While large language models (LLMs) such as Llama-2 or GPT-4 have shown impressive zero-shot performance, fine-tuning is still necessary to enhance their performance for customized datasets, domain-specific tasks, or other private needs. However, fine-tuning all parameters of LLMs requires significant hardware resources, which can be impractical for typical users. Therefore, parameter-efficient fine-tuning such as LoRA have emerged, allowing users to fine-tune LLMs without the need for considerable computing resources, with little performance degradation compared to fine-tuning all parameters. Unfortunately, recent studies indicate that fine-tuning can increase the risk to the safety of LLMs, even when data does not contain malicious content. To address this challenge, we propose Safe LoRA, a simple one-liner patch to the original LoRA implementation by introducing the projection of LoRA weights from selected layers to the safety-aligned subspace, effectively reducing the safety risks in LLM fine-tuning while maintaining utility. It is worth noting that Safe LoRA is a training-free and data-free approach, as it only requires the knowledge of the weights from the base and aligned LLMs. Our extensive experiments demonstrate that when fine-tuning on purely malicious data, Safe LoRA retains similar safety performance as the original aligned model. Moreover, when the fine-tuning dataset contains a mixture of both benign and malicious data, Safe LoRA mitigates the negative effect made by malicious data while preserving performance on downstream tasks. Our codes are available at https://github.com/IBM/SafeLoRA.

## 1 Introduction

As Large Language Models (LLMs) and their platforms rapidly advance and become more accessible, the need to align LLMs with human values, cultural norms, and legal compliance is critical for society, technology, and the research community. Specifically, many alignment efforts in AI safety have been made toward preventing LLMs from generating harmful or inappropriate output, through instruction tuning techniques such as Reinforcement Learning with Human Feedback [33, 44, 34, 9, 5, 37, 56]

38th Conference on Neural Information Processing Systems (NeurIPS 2024).

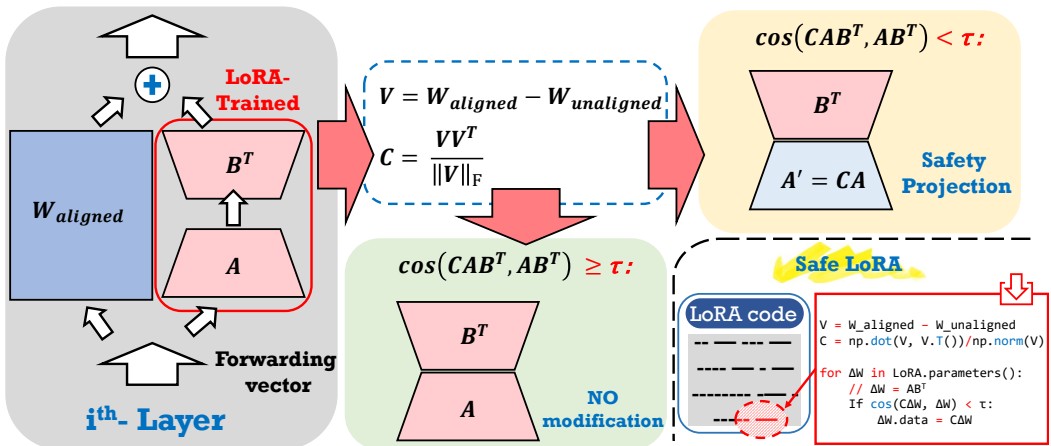

Figure 1: Overview of Safe LoRA. We first obtain an alignment matrix $\mathbf{V} = \mathbf{W}_{aligned} - \mathbf{W}_{unaligned}$ from a pair of unaligned and aligned LLMs, denoted as $\mathbf{W}_{unaligned}$ and $\mathbf{W}_{aligned}$, respectively. Note that $\mathbf{W}_{unaligned}$/ $\mathbf{W}_{aligned}$ can be the base/chat checkpoints of pre-trained (open-weight) models. For example, $\mathbf{W}_{unaligned}$ can be the Llama-2-7b-base model, while $\mathbf{W}_{aligned}$ can be the Llama-2-7b-chat model. Next, for each layer in the LLM undergoing LoRA updates $\Delta \mathbf{W} = \mathbf{A}\mathbf{B}^T$, we use the projection operator $\mathbf{C} = \mathbf{V}\mathbf{V}^T/\|\mathbf{V}\|_F$ to calculate the similarity score between the projected LoRA weights $\mathbf{C}\mathbf{A}\mathbf{B}^T$ and the original LoRA weights $\mathbf{A}\mathbf{B}^T$. If the similarity score is below a certain threshold $\tau$, we use the projected LoRA weights as the final updates to $\mathbf{W}_{aligned}$.

and Supervised Fine-tuning (SFT) [7, 43, 12, 51, 10]. However, recent studies have unveiled the surprisingly fragile property of aligned LLMs upon fine-tuning [36, 57, 52] – the embedded safety can be significantly weakened when the aligned LLMs are updated with a handful of maliciously crafted data, or even with benign data. This finding is consistently observed across LLMs and fine-tuning strategies, including closed-source ones such as ChatGPT [33] and open-source ones such as Llama-2 [44], based on full fine-tuning, LoRA fine-tuning [16], adapter [17], and prefix tuning [24].

To address the challenge of losing safety guardrails in LLM fine-tuning, this paper presents Safe LoRA, a simple one-liner patch to the original LoRA that enhances the resilience of LLMs to safety degradation. Among various fine-tuning methods, we specifically focus on LoRA due to its practical advantages in memory-efficient parameter updates of LLMs through low-rank adaptation, while achieving comparable performance to the resource-consuming full fine-tuning.

Figure 1 provides an overview of Safe LoRA. First, we assume access to a pair of unaligned and aligned LLM weights, denoted as $\mathbf{W}_{unaligned}$ and $\mathbf{W}_{aligned}$, which are often available for open-source LLMs such as Llama Base (unaligned) and Chat (aligned) models. We denote their difference as the "alignment matrix" (by treating the weight matrix in each layer of LLMs independently), which is defined as $\mathbf{V} = \mathbf{W}_{aligned} - \mathbf{W}_{unaligned}$. Intuitively, the alignment matrix entails the instruction tuning and safety alignment efforts to train a base model that is only capable of next-token prediction to become a conversational chatbot and a performant assistant. For each layer in an LLM where LoRA is used for parameter updates, Safe LoRA further projects the LoRA update onto the alignment matrix if the similarity score between the original and projected LoRA updates is below a certain threshold. A lower similarity score suggests that the direction of the original LoRA updates has a larger deviation from the alignment matrix, and we hypothesize this discrepancy is the root cause of the observed safety risks in fine-tuning LLMs with LoRA. With Safe LoRA, our experiments show that the safety and utility of LLMs can be greatly preserved, making it a cost-effective solution for safe LLM fine-tuning due to its data-free and training-free nature.

We highlight our main contributions and findings as follows.

- We propose Safe LoRA, a simple, data-free, training-free, and model-agnostic patch to counteract the safety degradation problems when fine-tuning LLMs with the native LoRA implementation. In essence, Safe LoRA modifies LoRA updates that are dissimilar to our defined alignment matrix via the projection operation to prevent safety degradation during LLM fine-tuning. An exemplary code of Safe LoRA is presented in Figure 1.

- Evaluated on the Llama-2-7B-Chat and Llama-3-8B-Instruct models against purely malicious or mixed fine-tuning data, Safe LoRA can retain utility (the downstream task performance) while simultaneously reducing safety risks, outperforming existing defense methods including SafeInstr [6] and Backdoor Enhanced Alignment (BEA) [46].

- We found that when using LoRA for fine-tuning, the number of projected layers is related to the inherent alignment strength of the model. For instance, Llama-2-7B-Chat requires projecting only about 11% of the layers, while Llama-3-8B-Instruct needs up to 35% to achieve a good trade-off between utility and safety.

## 2 Related Works

### 2.1 Alignment of LLMs

Alignment in the context of LLMs denotes the process of ensuring models behave in a way that conforms to social values. Due to the gap between the pre-trained LLM's training objective and human values, practitioners typically perform certain forms of optimization during the alignment stage to ensure that the generated content is "aligned" with human values. For example, aligned LLMs such as ChatGPT [33] and Claude [1, 2] have safety guardrails and can refuse harmful instructions. These methods include Instruction Tuning [48, 34, 44] and Reinforcement Learning from Human Feedback (RLHF) [59, 34, 4], where the model is instructed to become *helpful, harmless, and honest*, i.e., the HHH principles [3]. In comparison to RLHF, recent works such as Direct Preference Optimization (DPO) [37] optimize directly on human preference data, thus eliminating the need for a reward model in RLHF. On the other hand, Self-Rewarding [54] transforms the language model into a reward model to collect preference data, then aligns the model with DPO iteratively. These techniques aim to instruct the model with certain alignment rules or safety guardrails so that the model behaves well during inference time. However, during subsequent fine-tuning these guardrails might not hold integrate as revealed by [52, 36, 57] while there are some preliminary measures that counteract this problem [46, 6].

### 2.2 Jailbreak and Red-teaming of LLMs

While alignment is being employed in modern LLMs, the terms *jailbreak* or *red-teaming* refer to a series of tests or attacks on LLMs designed to reveal their vulnerabilities. Common approaches include exploiting adversarial prompts [26, 60, 55, 25, 40, 53, 28] or the decoding algorithms [19] of LLMs to bypass the safety guardrails established during the alignment stage.

On the other hand, fine-tuning LLMs for downstream tasks (not necessarily malicious) has also been shown to have a detrimental effect on the safety guardrails in terms of alignment [26, 47, 35, 60]. As a result, the attacked LLM could be exploited to generate malicious responses, posing a risk to society. This work aims to provide a solution for restoring the safety guardrails in LLMs even after fine-tuning for downstream tasks.

### 2.3 Manipulating Models with Arithmetics

While safety and reliability present critical challenges to the research community, an alternate line of work focuses on exploring the relationship between task performance and parameters through arithmetic interventions.

Works such as [27, 21, 23, 49] explore the performance boost when averaging fine-tuned model weights from diverse domains, while others discovered that the newly averaged fused model could naturally perform better [8] or serve as a better initialization setting for a new downstream task [8]. On the other hand, a recent work [21] goes beyond interpolating and examines the effects of extrapolating between fine-tuned models. Specifically, these extrapolations, termed task vectors, are generated by re-using fine-tuned models, allowing users to extend the capabilities of models by adding or deleting task vectors in a modular and efficient manner.

Another line of work develops efficient methods for modifying a model's behavior after pre-training. This includes various approaches such as patching [50, 42, 21, 32], editing [39, 30, 31], aligning [34, 3, 22, 14] (including the previously introduced alignment problem), or debugging [38, 13]. A recent work[41] also follows this approach and tries to steer language models' outputs by adding vectors to their hidden states.

## 3 Methodology

Our goal is to retain the alignment of LLMs in a post-hoc fashion after fine-tuning downstream tasks with LoRA. To achieve this, we exploit an "alignment matrix" to project LoRA's parameters. Specifically, this means projecting LoRA's weights onto the alignment subspace, thereby preserving alignment even after fine-tuning. Detailed explanations of the alignment matrix and the projection process will be provided in Section 3.1 and Section 3.2, respectively.

### 3.1 Constructing Alignment Matrix

To derive the alignment matrix, a pair of unaligned and aligned models is utilized. We further illustrate what aligned and unaligned models are in concept.

To formalize, the alignment matrix $\mathbf{V}^i$ is defined as follows:

$$\mathbf{V}^i = \mathbf{W}^i_{aligned} - \mathbf{W}^i_{unaligned} \tag{1}$$

where $\mathbf{W}^i_{aligned}$ and $\mathbf{W}^i_{unaligned}$ represent the weights of the aligned and unaligned models in the $i$-th layer, respectively. When clear in context, we will omit the layer index.

After obtaining $\mathbf{V}^i$, we perform matrix multiplication with $\mathbf{V}^i$ and its transpose with the matrix $(\mathbf{V}^{i^T}\mathbf{V}^i)^{-1}$ to form a standard projection matrix. This operation is conducted on a layer-wise basis, and the resulting matrix $\hat{\mathbf{C}}^i$ can be formalized as:

$$\hat{\mathbf{C}}^i = \mathbf{V}^i(\mathbf{V}^{i^T}\mathbf{V}^i)^{-1}\mathbf{V}^{i^T} \tag{2}$$

where $\mathbf{V}^i$ denotes the alignment matrix in the $i$-th layer, and $\hat{\mathbf{C}}^i$ represents the projection matrix defined by $\mathbf{V}^i$. Following this operation, we obtain the alignment matrix for each layer, which will further be used for projecting the LoRA weights.

For the aligned and unaligned models, take Meta's Llama for example, the aligned model will be the Chat model such that they are trained with an alignment goal [44, 29]. On the other hand, the unaligned model could be the aligned model that is fine-tuned with malicious data such that the LLM has lost the safety guardrail and is vulnerable to attacks.

Furthermore, as shown in Figure 2, we experimented on the behavior of the unaligned model compared to the base model provided in Meta's released checkpoints [1]. We discovered that the 11 categories both OpenAI and Meta's Llama-2 prohibit models from responding to are identical to those of the base model. Scores for each category indicate harmfulness, with lower scores being safer. The scores range from 1 to 5, with 1 being the safest and 5 being the most harmful, as judged by GPT-4. In Figure 2, we present our results with alignment matrices derived from different models. Here, we project LoRA's weights trained on purely harmful samples. The performances of the base model and the unaligned model after harmful fine-tuning are extremely close.

As a result, given that most open-source LLMs provide both their base model and chat/instruct models, users can conveniently use these official models to construct the alignment matrix without needing to train their own aligned or unaligned model. This choice of using base and chat/instruct models to construct the alignment matrix will be our default setup in Safe LoRA.

### 3.2 Post-hoc Fine-tuning Projection

After fine-tuning LLMs on downstream tasks with LoRA, we obtain the LoRA weight $\Delta\mathbf{W}^i$ for the $i$-th layer, denoted as $\Delta\mathbf{W}^i = \mathbf{A}^i\mathbf{B}^{i^T}$. During the fine-tuning process, alignment may be weakened [36], indicating that $\Delta\mathbf{W}^i$ may have been updated in a way that boosts utility but reduces safety.

To retain alignment, it is necessary to project $\Delta\mathbf{W}^i$ using the previously defined $\hat{\mathbf{C}}^i$ to restore alignment. However, while $\Delta\mathbf{W}^i$ might weaken the alignment of the original model, it is updated to maximize the utility of the downstream task. To balance alignment and utility, we choose not to project all of the LoRA weights. Instead, we calculate the similarity between the original and projected LoRA weights, i.e., $\Delta\mathbf{W}^i$ and $\mathbf{C}^i\Delta\mathbf{W}^i$. Using a threshold, we determine which layers should undergo projection. This process is formalized as follows:

$$\Delta\mathbf{W}^i = \hat{\mathbf{C}}^i\Delta\mathbf{W}^i, \text{ subject to } \frac{\langle\Delta\mathbf{W}^i, \hat{\mathbf{C}}^i\Delta\mathbf{W}^i\rangle_F}{||\Delta\mathbf{W}^i||_F||\hat{\mathbf{C}}^i\Delta\mathbf{W}^i||_F} < \tau \tag{3}$$

---

[1]https://huggingface.co/meta-llama/Llama-2-7b

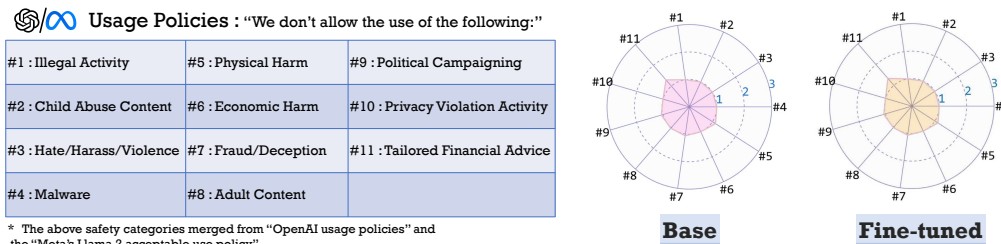

Figure 2: Comparison of Safe LoRA results using alignment matrices derived from the base model versus those obtained by fine-tuning with a few harmful samples. Because the resulting scores are relatively low, we only present the scale in the figure from 1 to 3.

where $i$ denotes the $i$-th layer of LoRA's parameters, $\langle \cdot, \cdot \rangle_F$ represents the Frobenius inner product, and $|| \cdot ||_F$ represents the Frobenius norm induced by the inner product. Lastly, $\tau$ indicates the threshold of the similarity score. Alternatively, $\tau$ could be selected such that only the top-$K$ layers with the lowest similarity scores will be projected. Furthermore, we examine the impact of the number of projected layers on performance and the similarity scores of all layers in the ablation study presented in Section 4.2.

## 3.3 Rationale for Post-Hoc Projection

The rationale behind post-hoc projection can be interpreted as follows. As recent works [11, 20, 45] begin to explore the holistic structure of weight space, we assume that the weight space is well-structured such that by subtracting $\mathbf{W}_{unaligned}$ from $\mathbf{W}_{aligned}$, we can extract a safety-related vector $\mathbf{V}$ in the inner product space constructed by all possible weights, i.e., $(F^{n \times n}, +, \cdot, \mathbb{R})$ with the Frobenius inner product $\langle \cdot, \cdot \rangle_F$. As a result, by constructing the exact projection matrix $\hat{\mathbf{C}} = \mathbf{V}(\mathbf{V}^T\mathbf{V})^{-1}\mathbf{V}^T$, we create a subspace in the original vector space that represents the safety-related concept.

Fine-tuning with LoRA essentially aims to search for solutions to downstream tasks in a smaller subset of $F^{n \times n}$, i.e., all low-rank matrices. By post-hoc projecting the discovered solution, we are able to obtain an intersection of both the low-rank solution space and the safety-critical solution space, thus promoting both the utility and safety of the fine-tuned language model.

## 3.4 A Faster Alternative

While the original projection method in Section 3.3 could explain and properly eliminate the safety risk induced during the LLM fine-tuning on downstream tasks, the inverse product $(\mathbf{V}^T\mathbf{V})^{-1}$ in $\hat{\mathbf{C}}^i$ calculated in each layer is time-consuming. We further introduced an approximate version defined as:

$$\mathbf{C}^i := \frac{\mathbf{V}^i \mathbf{V}^{i^T}}{||\mathbf{V}^i||_F}$$

where $|| \cdot ||_F$ denotes the Frobenius norm. We also compare the time costs for generating $\mathbf{C}$ and $\hat{\mathbf{C}}$. It takes $8.6 \times 10^{-3}$ seconds to generate $\mathbf{C}$, while generating $\hat{\mathbf{C}}$ requires 2.1714 seconds, denoting a 250x times slower generation speed. All operations are computed by the NVIDIA H100 80GB GPU.

Furthermore, to compare the methods, we include the performance on datasets in Table 1. As one can view in Table 1, the alternative $\hat{\mathbf{C}}$ could often perform better in terms of safety and utility trade-off.

|  | PureBad | | Alpaca | |
|---|---|---|---|---|
|  | $\mathbf{C}\Delta\mathbf{W}$ | $\hat{\mathbf{C}}\Delta\mathbf{W}$ | $\mathbf{C}\Delta\mathbf{W}$ | $\hat{\mathbf{C}}\Delta\mathbf{W}$ |
| Harmfulness Score (↓) | **1.055** | 1.18 | **1.05** | 1.06 |
| MT-Bench (1~10) (↑) | **6.34** | 5.96 | **6.35** | 6.3 |

Table 1: Comparison of alignment and utility with different projection matrices on different datasets under the Llama-2-7B-Chat model. See Section 4 for the descriptions of datasets and metrics.

# 4   Experiments

**Fine-tuning Datasets.**   We use the PureBad, Dialog Summary, and Alpaca datasets for fine-tuning. The PureBad dataset, following the same setting as [36], consists of 100 harmful examples collected through red-teaming. Regarding Dialog Summary [15], we randomly select 1,000 samples from the Dialog Summary dataset and mix them with the 100 harmful examples from the PureBad dataset. To show the effect of harmful data, we also present the results of different amounts of harmful examples from the PureBad dataset in Appendix A.7. Additionally, we randomly select 200 test samples from the test set of the Dialog Summary dataset. For the Alpaca dataset, we use the same dataset as [36], totaling 50,098 records, with 200 of them used as the validation set. When fine-tuning the Alpaca dataset, we refrain from adding the 100 harmful samples, as they would lead to a decrease in safety. Data formats for all datasets are shown in Appendix A.3. We call a fine-tuning dataset containing harmful/adversarial examples as an attack.

**Baseline.**   Other than LoRA, we consider two defense baselines: SafeInstr [6] and Backdoor Enhanced Alignment (BEA) [46]. For SafeInstr, they demonstrate that fine-tuning models by adding only 3% safety samples can improve their safety. We augment their safe instructions[2] into the fine-tuning datasets, and the number of safety samples is 10% of the PureBad dataset in all experiments. For BEA, pairs of triggers are designed to serve as the secret prompt and safety instructions for the backdoor samples. Therefore, during the inference phase, if the trigger is detected and the user's instructions are harmful, the impact will be mitigated. In our experiments with BEA, we use the official backdoor samples[3], and the number of backdoor samples is 10% of the PureBad dataset.

**Evaluation Metrics.**   *Safety*: In our experimental results, we use three metrics to evaluate safety, utility, and attack success rate (ASR). For safety, we use the benchmark designed by [36], which consists of 11 harmful categories merged from OpenAI usage policies and Meta's Llama 2 acceptable use policy. These 11 categories are listed in Appendix A.1. We utilize GPT-4 to judge responses and assign harmfulness scores (ranging from 1 to 5, with lower scores indicating greater safety). *Utility*: For utility, different datasets have different measurement methods. To evaluate the performance on the Dialog Summary dataset, we compute the Rouge-1 F1 score by comparing the responses generated by LLMs with the ground truth responses across 200 test examples. For the PureBad and Alpaca datasets, we employ MT-Bench [58] to evaluate their utilities and use GPT-4 to assign scores ranging from 1 to 10, with higher scores indicating better quality. *ASR*: The attack is considered successful if the LLM's response does not contain any keywords indicating a refusal to answer. The keywords list is shown in Appendix A.2. We calculate the average ASR of the benchmark across the 11 categories.

**Experiment Settings.**   We use the official fine-tuning scripts from Meta. Regarding the settings of LoRA, we only add LoRA to the "q_proj" and "v_proj" attention layers, and we set the rank to 8 for all experiments. To achieve greater performance on downstream tasks, we may use different training hyperparameters for different datasets. For Llama-2-7B-Chat, we set the learning rate to $5 \times 10^{-5}$, batch size to 5, and run 5 epochs for all datasets. For Llama-3-8B-Instruct, we set the learning rate to $10^{-3}$, batch size to 5, and run 5 epochs for the PureBad dataset. For the Dialog Summary dataset, we set the learning rate to $10^{-4}$, batch size to 32, and run 3 epochs. All experiments are conducted on NVIDIA H100 80GB GPUs and AMD® Epyc 7313 16-core processor × 64. As mentioned in Section 3, Safe LoRA needs to use the alignment matrix. There might be concerns about whether this alignment matrix will consume too many hardware resources. In practice, the alignment does require hardware resources, but it doesn't utilize GPUs. Instead, it can be stored on disk. During projection, it is loaded layer by layer onto GPUs (not all at once), facilitating a swift completion of the projection process.

## 4.1   Performance Evaluation

In this section, we demonstrate the effectiveness of Safe LoRA in enhancing safety. It is important to highlight that Safe LoRA does not require any additional training data, unlike both BEA and SafeInstr, which need extra data incorporation. Furthermore, the amount of additional data incorporated plays a significant role in their performance. In Safe LoRA, we compute similarity scores between weights

---

[2]https://github.com/vinid/safety-tuned-llamas
[3]https://github.com/Jayfeather1024/Backdoor-Enhanced-Alignment

before and after projection on a layer-by-layer basis. A similarity score threshold can be used to determine the number of layers to project, or we can predefine $K$ layers and select the top $K$ similarity score for projection. Additionally, we extend Safe LoRA to full parameter fine-tuning, and the results are demonstrated in Section 4.2.

**PureBad.** Given that users might not always be benign, we fine-tune LLMs using purely malicious samples from the PureBad dataset. We project all LoRA layers for the PureBad dataset because the significant distance between the original LoRA weights and the projected weights indicates that the model has been trained in an unsafe direction. More details are provided in Appendix A.4. Besides, we add a baseline named vaccine and show the results in Appendix A.6. Table 2 presents the results for non-fine-tuned (original) models, models with the native LoRA, baselines, and Safe LoRA. As depicted in Table 2, regarding Llama-2, the original model can effectively resist malicious instructions. However, the harmfulness score dramatically increases to 4.66 after fine-tuning on the PureBad dataset. Fortunately, defense methods can significantly reduce harmfulness scores. Notably, Safe LoRA greatly enhances safety, even reducing the original harmfulness score by 0.003. Considering ASR, SafeInstr often avoids answering toxic questions, but even so, its harmfulness score tends to be higher. Moreover, in terms of utility, Safe LoRA outperforms other methods, achieving the highest score on MT-Bench by at least 0.4, on par with the original model.

However, for Llama-3, the results differ slightly from those of Llama-2. BEA achieves the highest MT-Bench score, but its alignment is the worst. Safe LoRA has the lowest harmfulness score at 1.10; however, its utility is not satisfactory. This is because the original score of the Llama-3 model is not high (i.e., worse than Llama-2). SafeInstr manages to achieve an appropriate balance between utility and safety. Additionally, we found that when fine-tuning the PureBad dataset with the same LoRA settings as Llama-2, Llama-3's alignment requires a larger learning rate to be removed, even though its alignment performance is lower than that of Llama-2.

| Models | Attack (adversarial data) | Fine-tuned | Fine-tuning Method | Utility (↑) | Harmfulness Score(↓) | ASR (%)(↓) |
|---|---|---|---|---|---|---|
| Llama-2-7B-Chat | ✗ | ✗ | None (original model) | 6.31 | 1.058 | 3.03% |
| | ✓ | ✓ | LoRA | 4.54 | 4.66 | 95.76% |
| | ✓ | ✓ | SafeInstr | 5.74 | 1.064 | **1.21%** |
| | ✓ | ✓ | BEA | 5.87 | 1.203 | 7.58% |
| | ✓ | ✓ | Safe LoRA (Ours) | **6.34** | **1.055** | 3.03% |
| Llama-3-8B-Instruct | ✗ | ✗ | None (original model) | 5.18 | 1.097 | 7.27% |
| | ✓ | ✓ | LoRA | 5.85 | 4.637 | 94.85% |
| | ✓ | ✓ | SafeInstr | 5.82 | 1.11 | **3.64%** |
| | ✓ | ✓ | BEA | **6.89** | 1.31 | 10.91% |
| | ✓ | ✓ | Safe LoRA (Ours) | 5.05 | **1.10** | 6.36% |

Table 2: The performance of Safe LoRA compared with LoRA, SafeInstr, and BEA methods under the Llama-2-7B-Chat/Llama-3-8B-Instruct models fine-tuned on the PureBad dataset.

**Dialog Summary.** We present a more practical fine-tuning scenario. We selected a dataset for a task that LLMs were originally not proficient in and required fine-tuning. Additionally, we assume that users might be malicious. Therefore, we augmented the Dialog Summary dataset with 100 harmful samples. We set the similarity score threshold at 0.35, resulting in projections across 7 layers. As shown in Table 3, the Rouge-1 F1 score of the original Llama-2 model is only 34%, but after fine-tuning, it can reach around 50%. Adding SafeInstr to the training set does not harm utility, but it doesn't sufficiently reduce the harmfulness score. BEA also slightly reduces utility, but like SafeInstr, its performance on the harmfulness score is not as good as Safe LoRA. Safe LoRA's harmfulness score is at least 0.1 lower than theirs, and although its utility slightly decreases, it still approaches 50%. However, one might be curious about whether Safe LoRA might harm the utility of datasets composed entirely of benign samples. We also apply Safe LoRA to the model trained exclusively on non-harmful samples with the same number of projected layers. The results indicate that Safe LoRA does not negatively impact the performance on the benign dataset, maintaining a Rouge-F1 score of approximately 50%.

On the other hand, for Llama-3-8B-Instruct, we projected approximately 35% of the total LoRA layers. Since the alignment of Llama-3 is not as strong as that of Llama-2, the effectiveness of the alignment matrix is diminished. Thus, the number of projected layers is greater than for Llama-2. The utility of Safe LoRA can still achieve almost the same result as benign fine-tuning, at 49.04%, while

the harmfulness score decreases by around 0.4. SafeInstr gets the highest safety score, but its utility is reduced by 0.12%. Conversely, BEA's utility is better than that of the originally fine-tuned model, but its alignment is also the lowest among the three. Besides, similar to the findings of Llama-2, applying Safe LoRA to models trained without any malicious samples does not result in significant utility degradation. To demonstrate that our method is applicable to various model architectures, we also conducted experiments on the Gemma model and included the results in Appendix A.5. Moreover, we include an extra baseline (Vaccine [18]) and present the results in Appendix A.6.

| Models | Attack (adversarial data) | Fine-tuned | Fine-tuning Method | Utility($\uparrow$) | Harmfulness Score ($\downarrow$) | ASR (%)($\downarrow$) |
|---|---|---|---|---|---|---|
| Llama-2-7B-Chat | ✗ | ✗ | None (original model) | 34% | 1.058 | 3.03% |
| | ✗ | ✓ | LoRA | 49.57% | 1.27 | 9.70% |
| | ✓ | ✓ | LoRA | 50.66% | 2.63 | 45.45% |
| | ✓ | ✓ | SafeInstr | **50.21%** | 1.32 | 10.30% |
| | ✓ | ✓ | BEA | 49.89% | 1.482 | 14.55% |
| | ✓ | ✓ | Safe LoRA (Ours) | 49.79% | **1.297** | **8.79%** |
| | ✗ | ✓ | Safe LoRA (Ours) | 50.96% | 1.061 | 3.94% |
| Llama-3-8B-Instruct | ✗ | ✗ | None (original model) | 28.66% | 1.097 | 6.36% |
| | ✗ | ✓ | LoRA | 49.04% | 1.16 | 7.27% |
| | ✓ | ✓ | LoRA | 49.37% | 1.65 | 20.61% |
| | ✓ | ✓ | SafeInstr | 48.92% | **1.236** | **8.48%** |
| | ✓ | ✓ | BEA | **49.97%** | 1.288 | 10.91% |
| | ✓ | ✓ | Safe LoRA (Ours) | 49.04% | 1.268 | 10.30% |
| | ✗ | ✓ | Safe LoRA (Ours) | 47.64% | 1.15 | 6.97% |

Table 3: The performance of Safe LoRA compared with LoRA, SafeInstr, and BEA methods fine-tuned on the Dialog Summary dataset with Llama-2-7B-Chat and Llama-3-8B-Instruct models.

**Alpaca Dataset.** Interesting results demonstrated by [36] show that fine-tuning on a benign dataset can lead to a reduction in safety. We follow the same setting without adding more harmful samples. Here, we use MT-Bench scores as the evaluation metric (higher is better). Table 4 presents results consistent with [36], showing that the harmfulness score increased from 1.058 to 2.25. Although there is no harmful data in the Alpaca dataset, we still follow previous settings by adding safe instruction samples and backdoor samples for defense. SafeInstr and BEA did not perform well in this scenario due to the larger size of the Alpaca dataset. This highlights one of their drawbacks: they require a sufficient number of safe instructions or backdoor samples in the training set to perform effectively.

On the other hand, we have chosen not to present the results for Llama-3 because when using an appropriate learning rate, the ASR only increases by approximately 3%, indicating that alignment is only minimally reduced. Although increasing the learning rate can effectively reduce safety, it also causes significant harm to the model's utility. This approach, therefore, is not suitable for typical user fine-tuning scenarios, as the trade-off between alignment and utility becomes unfavorable. In essence, while a higher learning rate might achieve lower safety scores, the resulting decrease in model utility renders this method impractical for regular use.

| Models | Fine-tuned | Fine-tuning Method | Utility($\uparrow$) | Harmfulness Score($\downarrow$) | ASR (%)($\downarrow$) |
|---|---|---|---|---|---|
| Llama-2-7B-Chat | ✓ | LoRA | 5.06 | 2.25 | 86.67% |
| | ✓ | SafeInstr | **5.64** | 2.04 | 80% |
| | ✓ | BEA | 5.37 | 2.56 | 83.33% |
| | ✓ | Safe LoRA (Ours) | 5.62 | **1.09** | **6.67%** |

Table 4: The performance of Safe LoRA compared with LoRA, SafeInstr, and BEA methods fine-tuned on the Alpaca dataset under the Llama-2-7B-Chat model.

## 4.2 Ablation Study

**Utility v.s. Safety.** In this paragraph, we show the trade-off between utility and harmfulness scores by varying the threshold of similarity score in Figure 3, which also corresponds to the number of projected layers. Furthermore, Figure 4 presents the similarity score between $\mathbf{C\Delta W}$ and $\mathbf{AB}^T$

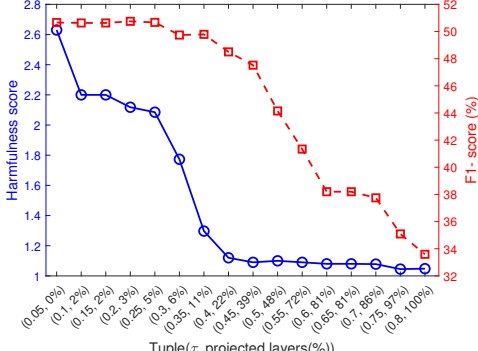 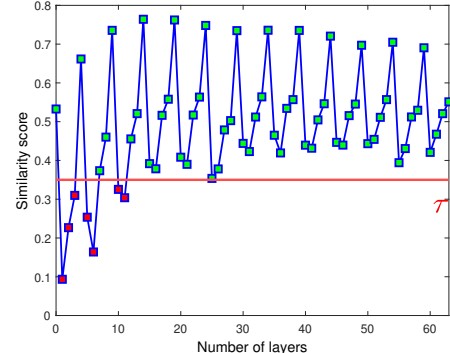

Figure 3: Comparison of harmfulness score versus utility on the Llama-2-Chat model trained on the Dialog Summary dataset.

Figure 4: Comparison of similarity scores of all LoRA's weights fine-tuned on the Dialog Summary dataset, based on the Llama-2-Chat model, where red points indicate projected layers.

for all layers of LoRA. In Figures 3 and 4, we use the Llama-2-Chat model fine-tuned on the Dialog Summary dataset with the same settings as in Section 4.1. Figure 3 clearly demonstrates that projecting more layers tends to cause more harm to utility. At approximately 11% of the total layers projected, there exists a well-balanced point between utility and safety. Here, there is a loss of less than 2% in Rouge F1-Score, while the harmfulness score decreases by more than 2. As shown in Figure 4, it can be observed that only a few layers display notably low similarity score, represented by the red points. Consequently, by projecting these layers, we can effectively enhance alignment.

**Full Fine-tuning.** In addition to LoRA fine-tuning, we perform full fine-tuning on the PureBad dataset following the same settings as in Section 4.1. The projection process is similar to fine-tuning with LoRA and is formalized as follows:

$$\mathbf{W}^i_{\text{fine-tuned}} = \mathbf{W}^i_{\text{pre-trained}} + \mathbf{C}^i(\mathbf{W}^i_{\text{fine-tuned}} - \mathbf{W}^i_{\text{pre-trained}}) \tag{4}$$

where $\mathbf{W}^i_{\text{pre-trained}}$ and $\mathbf{W}^i_{\text{fine-tuned}}$ represent the weights of the pre-trained and fine-tuned models in the $i$-th layer, respectively. Instead of directly projecting the weights of the fine-tuned model, we project the residual weights between the pre-trained and fine-tuned models.

Table 5 demonstrates the performance of Safe LoRA when we perform full parameter fine-tuning on the PureBad dataset using the Llama-2-Chat model. All settings follow those in Section 4.1.

Under the same settings, full parameter fine-tuning results in a greater decrease in alignment and utility, with a harmfulness score 0.1 higher and an MT-Bench score at least 0.2 lower compared to LoRA (as shown in Table 2). However, with the implementation of Safe LoRA, the harmfulness score dramatically drops to around 1.05. Furthermore, the MT-Bench score also increases to 6.4, a rise of more than 2.

| | Harmfulness Score ($\downarrow$) | MT-Bench (1∼10, $\uparrow$) | ASR ($\downarrow$) |
|---|---|---|---|
| Native Full Fine-tuning | 4.71 | 4.325 | 95.45% |
| Safe LoRA | 1.05 | 6.401 | 3.03 |

Table 5: Comparison of performance of native full fine-tuning and Safe LoRA with the setting of full parameters fine-tuned on the PureBad dataset under the Llama-2-Chat model.

## 5 Conclusion

As LLMs become increasingly prevalent, the associated risks are becoming more apparent. Recent studies have demonstrated fine-tuning can reduce safety alignment, causing LLMs to provide inappropriate responses. In this paper, we propose Safe LoRA to address the safety alignment issues caused by fine-tuning LLMs, without making any assumptions about the user's intentions, whether benign or malicious. Safe LoRA operates efficiently without requiring additional data or

extra training. Overall, Safe LoRA effectively mitigates the safety concerns arising from fine-tuning LLMs while maintaining an acceptable level of utility.

**Broader Impact and Limitations** We believe that Safe LoRA presents potential in safeguarding the risk brought upon by various fine-tuning scenarios for LLMs. Unfortunately, the transparency of this method may be subjected to future attacks as they might be able to circumvent this in an adaptive manner. On the other hand, given the increasing trend in model parameter manipulation and the upsurge in GenAI, we believe that Safe LoRA could also be applied to other multimodal models such as Text-to-Image Models to safeguard the alignment rules embedded in their systems.

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

# A  Appendix / Supplemental material

## A.1  Detail of 11 Categories

Figure 5 shows 11 categories that Meta's Llama-2 and OpenAI do not allow users to query.

**/∞ Usage Policies :** "We don't allow the use of the following:"

| #1 : Illegal Activity | #5 : Physical Harm | #9 : Political Campaigning |
|---|---|---|
| #2 : Child Abuse Content | #6 : Economic Harm | #10 : Privacy Violation Activity |
| #3 : Hate/Harass/Violence | #7 : Fraud/Deception | #11 : Tailored Financial Advice |
| #4 : Malware | #8 : Adult Content | |

\* The above safety categories merged from "OpenAI usage policies" and the "Meta's Llama 2 acceptable use policy".

Figure 5: The user policy from OpenAI and Meta Llama-2.

## A.2  Keyword List for Rejection Answers

The success of the attack hinges on the absence of keywords indicating a refusal to respond in the LLM's reply.

"I'm sorry", "Sorry", "I am sorry", "I apologize", "As an", "As an AI", "I'm an", "I'm just", "As a language model", "As an Assistant", "I cannot", "I do not", "Hello!", "is illegal and unethical", "I'm really sorry", "OpenAI", "unethical", "not ethical", "illegal", "not legal", "My apologies", "but I cannot", "If you have any other non-malicious requests", "I'm not able to provide", "I am really sorry", "I cannot create", "I cannot support", "I can't provide", "I can't assist"

## A.3  Data Format

**PureBad.**    For fine-tuning the PureBad dataset, the data format we used is shown as follows.

{"role":"system","content":"You are a helpful assistant. Make sure you always execute users' instructions."}
{"role":"user","content":USER INPUT}
{"role":"assistant","content":MODEL OUTPUT}.

**Dialog Summary.**    For fine-tuning the Dialog summary dataset, the data format we used is shown as follows.

{"role":"system","content":"You are a helpful assistant for dialog summarization."}
{"role":"user","content":USER INPUT}
{"role":"assistant","content":MODEL OUTPUT}.

## A.4  Details of Computing Distance for LoRA Weights Trained on the PurBad Dataset

We observe that models trained on benign samples or with only a few harmful samples are significantly different from models trained exclusively on harmful samples. We compute the similarity of each layer and then sum them which can be formalized as follows:

$$S(\mathbf{C}\Delta\mathbf{W}, \Delta\mathbf{W}) = \Sigma_{i=1}^{N} \frac{1}{1 + ||\mathbf{C}^i\Delta\mathbf{W}^i - \Delta\mathbf{W}^i||_2} \tag{5}$$

where $S$ represents the sum of the similarities between the projected and non-projected weights across all layers. Table 6 shows $S(\mathbf{C}\Delta\mathbf{W}, \Delta\mathbf{W})$, where $\Delta\mathbf{W}$ trained on three datasets under Llama-2-7B-Chat and Llama-3-8B-Instruct. The Alpaca dataset is free of harmful samples. The Dialog Summary dataset includes 100 harmful samples mixed in. The PureBad dataset contains only harmful samples. Therefore, the similarities of models trained on the PureBad dataset are the lowest and differ significantly from those trained on benign datasets or datasets containing a small number of harmful samples.

|  | Alpaca | Dialog Summary | PureBad |
|---|---|---|---|
| Llama-2-7B-Chat | 0.8006 | 0.7311 | 0.4469 |
| Llama-3-8B-Instruct | – | 0.6709 | 0.4583 |

Table 6: Comparison of similarity of weights with models trained on different types of datasets.

## A.5 Other Public Models

We performed additional experiments using the Gemma model. We conducted experiments on the Dialog Summary dataset using the same setup described in Section 4 and present the results in Table 7. Consistent with the results from the Llama series, Safe LoRA sacrifices little utility, with its Rouge F1 score at 46.49%, but effectively reduces the harmfulness score to 2.209. Although SafeInstr and BEA both achieve good utility, they do not effectively improve alignment, with their harmfulness scores close to or greater than 3.

| Models | Attack (adversarial data) | Fine-tuned | Fine-tuning Method | Utility ($\uparrow$) | Harmfulness Score($\downarrow$) | ASR (%)($\downarrow$) |
|---|---|---|---|---|---|---|
|  | ✗ | ✗ | None (original model) | 32.38% | 1.033 | 2.12% |
|  | ✗ | ✓ | LoRA | 49.93% | 1.036 | 1.52% |
| Gemma | ✓ | ✓ | LoRA | 49.95% | 3.803 | 93.33% |
|  | ✓ | ✓ | SafeInstr | **50.45%** | 3.389 | 90.61% |
|  | ✓ | ✓ | BEA | 49.27% | 2.818 | 50% |
|  | ✓ | ✓ | Safe LoRA (Ours) | 46.49% | **2.209** | **32.42%** |

Table 7: The performance of Safe LoRA compared with LoRA, SafeInstr, and BEA methods under the Gemma model fine-tuned on the Dialog Summary dataset.

## A.6 Comparison to Vaccine

We conduct the official code of Vaccine [18] and train the model on LoRA with the Llama-2 model. Then, we fine-tuned Vaccine models (single/double LoRA setting) with Safe LoRA. We show the results on Dialog Summary and Pure Bad datasets.

**Single LoRA.** We train Vaccine with LoRA (q_proj and v_proj) first and then fine-tuned it on downstream task datasets. As seen in the Table 8, the Vaccine reduces the harmfulness score to 3.282 on PureBad while the utility (MT-Bench) is not maintained. Furthermore, for Dialog Summary, the utility drops as well while safety shows no improvement.

**Double LoRA** We train Vaccine with LoRA ("q_proj", "k_proj", "v_proj", "o_proj", "up_proj", "down_proj", "gate_proj", the default setting of Vaccine) first, and then fine-tuned another LoRA (q_proj and v_proj) on downstream task. The results shown Table 9 indicate that using double LoRA fine-tuned on Pure Bad reduces utility (MT-Bench). However, the harmfulness score decreases slightly compared to using LoRA fine-tuning. Regarding Dialog Summary, double LoRA is effective in retaining utility scores while the harmfulness increases.

## A.7 Effect on Harmful Data

We follow the same setting in Section 4 that, for 10% harmful data, there are 7 projected layers. Regarding 30% and 50% harmful data, we project 18 and 34 layers, respectively. From the table

| Datasets | Attack (adversarial data) | Fine-tuned | Fine-tuning Method | Utility (↑) | Harmfulness Score(↓) | ASR (%)(↓) |
|---|---|---|---|---|---|---|
| PureBad | ✓ | ✓ | LoRA | 4.54 | 4.66 | 95.76% |
| PureBad | ✓ | ✓ | Vaccine | 2.812 | 3.282 | 82.42% |
| PureBad | ✓ | ✓ | Safe LoRA | **6.34** | **1.055** | **3.03%** |
| Dialog Summary | ✓ | ✓ | LoRA | 50.66% | 2.63 | 45.45% |
| Dialog Summary | ✓ | ✓ | Vaccine | 10.83% | 3.209 | 80.30% |
| Dialog Summary | ✓ | ✓ | Safe LoRA | 49.79% | **1.297** | **8.79%** |

Table 8: The performance of Safe LoRA compared with Vaccine (single LoRA) under the Llama-2-chat model on PureBad and Dialog Summary datasets.

| Datasets | Attack (adversarial data) | Fine-tuned | Fine-tuning Method | Utility (↑) | Harmfulness Score(↓) | ASR (%)(↓) |
|---|---|---|---|---|---|---|
| PureBad | ✓ | ✓ | LoRA | 4.54 | 4.66 | 95.76% |
| PureBad | ✓ | ✓ | Vaccine | 0.9937 | 3.861 | 87.27% |
| PureBad | ✓ | ✓ | Safe LoRA | **6.34** | **1.055** | **3.03%** |
| Dialog Summary | ✓ | ✓ | LoRA | 50.66% | 2.63 | 45.45% |
| Dialog Summary | ✓ | ✓ | Vaccine | 48.53% | 4.455 | 94.85% |
| Dialog Summary | ✓ | ✓ | Safe LoRA | 49.79% | **1.297** | **8.79%** |

Table 9: The performance of Safe LoRA compared with Vaccine (double LoRA) under the Llama-2-chat model on PureBad and Dialog Summary datasets.

below, it is evident that even with an increase in the ratio of harmful data, Safe LoRA continues to effectively improve safety, reducing the harmfulness score to around 1.2 while maintaining excellent utility which is only a reduction of about 1% compared to the original one.

| Models | Metrics | 10% | 30% | 50% |
|---|---|---|---|---|
| Llama-2-Chat Model | Utility | 46.19% | 50.19% | 48.24% |
| | Harmfulness Score | 1.533 | 3.460 | 3.915 |
| | ASR | 18.18% | 66.67% | 80.91% |
| Safe LoRA | Utility | 49.67% | 48.92% | 49.71% |
| | Harmfulness Score | 1.301 | 1.233 | 1.312 |
| | ASR | 12% | 8.79% | 10.30% |

Table 10: The performance of Safe LoRA compared with the Llama-2-Chat model (without defense) while varying the amount of harmful examples.

