# OpenReview forum: "Safe LoRA: The Silver Lining of Reducing Safety Risks when Finetuning Large Language Models"
_NeurIPS.cc/2024/Conference — NeurIPS 2024 poster_

### Official Review · Reviewer_X4Co · 2024-06-13

**Soundness:** 3
**Presentation:** 3
**Contribution:** 3
**Rating:** 7
**Confidence:** 4

**Summary:**

This paper proposes Safe LoRA to defend against the harmful finetuning issue for LLMs. The core idea of Safe Lora is to project the harmful gradient update to the subspace constructed by the alignment update.  To guarantee utility performance, the authors propose to use cosine similarity to determine whether the update in a layer should be projected or not.

**Strengths:**

1. This paper proposes a timely solution for the harmful finetuning issue for LLMs.

2. The idea is intuitive enough and should be an effective solution to the problem.

3. The refinement that uses cosine similarity to decide whether the projection is done for each layer is interesting, and brings practical performance enhancement.

4. The paper is well-written and concise enough, and I think the potential audience of this paper will be large given its simplicity.

**Weaknesses:**

1) Baseline selection can be more comprehensive. While the authors compare with  SafeInstr and BEA, the authors might also consider comparing with Vaccine (Huang et al, 2024), which is first available earlier than BEA and with source code available.

Huang T, Hu S, Liu L. Vaccine: Perturbation-aware alignment for large language model[J]. arXiv preprint arXiv:2402.01109, 2024.

2) Some important literature is missing for the discussion. Please consider reviewing these related papers on harmful finetuning.

[1] Fine-tuning can cripple your foundation model; preserving features may be the solution https://openreview.net/forum?id=VQ7Q6qdp0P (ICLR2024 template)

[2] Vaccine: Perturbation-aware Alignment for Large Language Model    https://arxiv.org/abs/2402.01109  （ICML2024 template）

[3] Keeping LLMs Aligned After Fine-tuning: The Crucial Role of Prompt Templates https://arxiv.org/pdf/2402.18540 （ACL2024 template）

[4] Immunization against harmful fine-tuning attacks https://arxiv.org/pdf/2402.16382 （ICLR2024 workshop template）

[5] Safety Fine-Tuning at (Almost) No Cost: A Baseline for Vision Large Language Models https://arxiv.org/pdf/2402.02207   （ICML2024 template）

------------------------------------------------------------Concurrent------------------------------------------------------------

[6] Representation noising effectively prevents harmful fine-tuning on LLMs   https://arxiv.org/pdf/2405.14577 （NeurIPS2024 template）

[7] Lazy Safety Alignment for Large Language Models against Harmful Fine-tuning   https://arxiv.org/abs/2405.18641   （NeurIPS2024 template）

[8] No Two Devils Alike: Unveiling Distinct Mechanisms of Fine-tuning Attacks   https://arxiv.org/pdf/2405.16229 （NeurIPS2024 template）

[9] A safety realignment framework via subspace-oriented model fusion for large language models https://arxiv.org/pdf/2405.09055 （also a post-hoc defense like safelora)

[10] Navigating the Safety Landscape: Measuring Risks in Finetuning Large Language Models https://arxiv.org/abs/2405.17374 （NeurIPS2024 template）

I am aware that some of the listed work is concurrent work (e.g., concurrent submissions to NeurIPS 2024). However, it is encouraged to also cite and discuss them, because that will be beneficial for the development of the research field (but the authors should at least cite and discuss those existing works that appeared before the NeurIPS2024 review cycle).

3. The experiment can be more comprehensive. For example, will the ratio of harmful data affect the defense performance?

**Questions:**

How do you construct the alignment update in the experiment? In section 3.1, it is claimed that:

> For the aligned and unaligned models, take Meta’s Llama for example, the aligned model will be the Chat model such that they are trained with an alignment goal [43, 28]. On the other hand, the unaligned model could be the aligned model that is fine-tuned with malicious data such that the LLM has lost the safety guardrail and is vulnerable to attacks.

I am wondering in the experiments, are you using harmful data to first unaligned the chat model, and then obtain the alignment update?

If this is the case, I am wondering if the safe gradient update can also be obtained by these procedures.  i) Use vanilla Llama2 (not chating version) as an unaligned model. ii) Do the alignment on Llama2 and get the aligned model.  iii) Subtract the weights of these two models.

**Limitations:**

The paper discusses the limitations. I don't think the mentioned limitation should be a problem in the acceptance of this paper. I am willing to further increase the score if the authors can provide experiments to address my raised concerns.

---

> ### Author Rebuttal · Authors · 2024-08-07
>
> We sincerely thank the reviewer for offering detailed reviews on$\textsf{Safe LoRA}$. We appreciated the comment saying that “The paper is well-written and concise enough, …  the potential audience of this paper will be large given its simplicity.” and shared a common perspective as the reviewer, laying importance in restoring alignment under practical scenarios. Below we provide a pointwise reply to the reviewer’s comment.
>
> &nbsp;
>
> **Literature Discussion**
>
> We thank the reviewer for bringing up related works and concurrent work for discussion. We would cite and discuss the mentioned papers in the revised version. For a brief introduction, please see the general response for the literature discussion.
>
> &nbsp;
>
> **More Performance Comparison**
>
> We conduct the official code of Vaccine and train the model on LoRA with the Llama-2 model. Then, we fine-tuned Vaccine models (single/double LoRA setting) with $\rho=2$. We show the results on Dialog Summary and Pure Bad datasets.
>
> **Single LoRA**
>
> We train Vaccine with LoRA (q_proj and v_proj) first, and then fine-tuned it on downstream task datasets. As seen in the table below, Vaccine reduces the harmfulness score to 3.282 on PureBad while the utility (MT-Bench $\uparrow$) is not maintained. Furthermore, for Dialog Summary, the utility drops as well while safety shows no improvement.
>
> |    Dataset     | Attack(adversarial data) | fine-tuned | Fine-tuning Method | Utility | Harmfulness Score |  ASR   |
> |:--------------:|:----------------------------:|:---------:|:------------------:|:-------:|:-----------------:|:------:|
> |    Pure Bad    |              ✓               |     ✓     |        LoRA        |  4.54   |       4.66        | 95.76% |
> |    Pure Bad    |              ✓               |     ✓     |      Vaccine       |  2.812  |       3.282       | 82.42% |
> |  Dialog Summary              |     ✓                         |     ✓      |     LoRA               |   50.66%      |     2.63              |  45.45%      |
> | Dialog Summary |              ✓               |     ✓     |      Vaccine       | 10.83%  |       3.209       | 80.30% |
>
> &nbsp;
>
> **Double LoRA**
>
> We train Vaccine with LoRA ("q_proj", "k_proj", "v_proj", "o_proj", "up_proj", "down_proj", "gate_proj", default setting of Vaccine) first, and then fine-tuned another LoRA (q_proj and v_proj) on downstream task. The results shown below indicate that using double LoRA fine-tuned on Pure Bad reduces utility (MT-Bench $\uparrow$). However, the harmfulness score decreases slightly compared to using LoRA fine-tuning. Regarding Dialog Summary, double LoRA is effective in retaining utility scores while the harmfulness increases.
>
>
> |    Dataset     | Attack(adversarial data) | fine-tuned | Fine-tuning Method | Utility | Harmfulness Score |  ASR  |
> |:--------------:|:----------------------------:|:---------:|:------------------:|:-------:|:-----------------:|:-----:|
> |    Pure Bad    |              ✓               |     ✓     |        LoRA        |  4.54   |       4.66        | 95.76% |
> |    Pure Bad    |              ✓               |     ✓     |      Vaccine       |   0.9937|   3.861       |87.27% |
> |  Dialog Summary              |     ✓                         |     ✓      |     LoRA               |   50.66%      |     2.63              |  45.45%      |
> | Dialog Summary |              ✓               |     ✓     |      Vaccine       | 48.53%  |   4.455      |94.85% |
>
> Although it seems that Vaccine does not effectively reduce harmfulness, it might be contributed to the fact that Alpaca was used for vaccine models in which the alignment might be compromised as shown in [R6]. Due to the time limit, we are unable to validate the cause which requires multiple experiments. However, we deem that both $\textsf{Safe LoRA}$ and Vaccine are diverse viable solutions toward soothing the misalignment of LLM fine-tuning and should both be considered during practical deployment.
>
> &nbsp;
>
> [R6]: Xiangyu Qi, Yi Zeng, Tinghao Xie, Pin-Yu Chen, Ruoxi Jia, Prateek Mittal, and Peter Henderson. Fine-tuning aligned language models compromises safety, even when users do not intend to! In The Twelfth International Conference on Learning Representations (ICLR 2024).
>
> &nbsp;
>
> **Effect on Harmful Data**
>
> We thank the reviewer for this additional question on experiments which we answer below. We follow the same setting in Section 4 that, for 10% harmful data, there are 7 projected layers. Regarding 30% and 50% harmful data, we project 18 and 34 layers, respectively.
> From the table below, it is evident that even with an increase in the ratio of harmful data, $\textsf{Safe LoRA}$ continues to effectively improve safety, reducing the harmfulness score to around 1.2 while maintaining excellent utility which is only a reduction of about 1% compared to the original one.
>
> |  Original Model   |  10%   |  30%   |  50%   |
> |:-----------------:|:------:|:------:|:------:|
> |      Utility      | 49.16% | 50.19% | 48.24% |
> | Harmfulness Score | 1.533  | 3.460  | 3.915  |
> |        ASR        | 18.18% | 66.67% | 80.91% |
>
> |      $\textsf{Safe LoRA}$      |  10%   |  30%   | 50% |
> |:-----------------:|:------:|:------:|:---:|
> |      Utility      | 49.67% | 48.92% | 49.71%    |
> | Harmfulness Score | 1.301  |   1.233     |   1.312  |
> |        ASR        |  12%   | 8.79%  | 10.30%    |
>
> &nbsp;
>
> **Building Alignment Update**
>
> The reviewer is exactly correct in the way of building alignment updates! In fact, we’ve written that in section 3.1, the base model is eventually considered as it shows similar performance to the maliciously fine-tuned one. Here we quote “As a result, … most open-source LLMs provide both their base model and chat/instruct models, users can conveniently use these official models to construct the alignment matrix without needing to train their own aligned or unaligned model.” To sum up, we note that here $\textsf{Safe LoRA}$ is a training-free and data-independent method as it requires only the aligned and base model.

---

> > ### Comment · Reviewer_X4Co · 2024-08-07
> > **Thanks for the rebuttal**
> >
> > Thanks for the rebuttal. I am not sure how to buid alignment update in **your experiment**. Which way you are using?
> >
> > 1. Use harmful data to first unaligned the chat model, and then obtain the alignment update.
> >
> > 2. Use vanilla Llama2 (not chating version) as an unaligned model. ii) Do the alignment on Llama2 and get the aligned model. iii) Subtract the weights of these two models.

---

> ### Author Response · Authors · 2024-08-07
> **Clarification of Alignment Update**
>
> We thank the reviewer for the additional question which we will try to clarify as the following. In Section 3.1, we explain $\textsf{Safe LoRA}$ by constructing the alignment update with the unaligned model, i.e., alignment update = chat - unaligned. However, as mentioned in the rebuttal and original paper “*We discovered that … are identical to those of the base model*”. The base model is eventually considered to construct the alignment update in $\textsf{Safe LoRA}$, i.e., **alignment update = chat - base and all subsequent experiments (Table 1 through 5), as it is a more practical scenario**.

---

> ### Comment · Reviewer_X4Co · 2024-08-07
> **Some questions on comparison with Vaccine**
>
> Thanks for the prompt answer of constructing alignment update, I think the authors can modify the equation to showcase this as this is a very important procedure. I am kind of confused about this point when reading your paper.
>
> I still have a few questions regarding the comparison with Vaccine:
>
> 1. are you using chat model (aligned model) as the base model for Vaccine?
> 2. what alignment dataset you are using to train Vaccine?

---

> > ### Author Response · Authors · 2024-08-07
> > **More details about Vaccine**
> >
> > Thank you for the reviewer's suggestion. We will emphasize after equation (1) that, in practice, we use the base model as the unaligned model, so users do not need additional data and training processes to obtain an unaligned model.
> >
> > &nbsp;
> >
> > We also appreciate the reviewer's additional question regarding the Vaccine implementation. We used the **Llama-2 Chat model** as the base model for Vaccine and followed the official code, which utilizes **Alpaca** as the alignment dataset.

---

> ### Comment · Reviewer_X4Co · 2024-08-07
> **On Vaccine implementation**
>
> Hi, thanks for the quick answer. If I got it correctly, in their official code, the alignment dataset used by Vaccine is not Alpaca, but a harmful prompt-safe answer dataset named BeaverTails. Aligned the model with Alpaca is not a correct way.
>
> I hope the  authors can address this issue, as a fair evaluation with baselines is our important to the value of the work.

---

> > ### Author Response · Authors · 2024-08-07
> > **Clarification of Vaccine Implementation**
> >
> > Thank you for the reviewer's feedback. You are correct that Alpaca is not an alignment dataset; we misunderstood its role. However, we used the official code provided [R1] and ran "*Vaccine.sh*" **without any modifications** to train Vaccine. Thus, the **alignment dataset we used is indeed BeaverTails_safe**. Vaccine mixed the Alpaca dataset with BeaverTails to prevent potential performance degradation that could occur from using only BeaverTails, thus incorporating normal data with Alpaca.
> >
> > &nbsp;
> >
> > [R1] https://github.com/git-disl/Vaccine/blob/main/script/alignment/Vaccine.sh

---

> ### Comment · Reviewer_X4Co · 2024-08-08
> **Thanks for the rebuttal. I have increased my score**
>
> Thanks for the clarification. Vaccine fails in some of your experiments probably because adding perturbation might break the original alignment of the Llama2-chat model (I think originally they used Llama2 (unaligned) model as base model).  Safe Lora does not have that issue because it is a post-fine-tuning solution. That said, I still encourage the authors to include the comparison results with Vaccine into their next version of the paper, as they show a new observation and also show the superiority of the Safe Lora method.
>
> I appreciate the author's willingness to provide additional experiments/baselines to enrich their work.  I therefore increase my score to 7. I will also actively participate the reviewer-AC discussion phase to support this paper.

---

> > ### Author Response · Authors · 2024-08-09
> > **Thanks for the Feedback**
> >
> > We appreciate the reviewer's active engagement in the discussion at this stage and the decision to raise the score. It is indeed possible that adding perturbations to an aligned model could disrupt the alignment, potentially making Vaccine's results less effective. However, Vaccine represents an initial but important solution to the realignment problem of LLMs. We will also include the results of Vaccine in the next version of our paper.

---

### Official Review · Reviewer_d4Du · 2024-07-05

**Soundness:** 3
**Presentation:** 3
**Contribution:** 3
**Rating:** 7
**Confidence:** 4

**Summary:**

This paper studies the problem that finetuning may compromise safety, as observed in the previous work. The author proposes Safe LoRA, a simple one-liner patch to the original LoRA implementation by introducing the projection of LoRA weights from selected layers to the safety-aligned subspace, reducing the safety risks in LLM fine-tuning while maintaining utility. Safe Lora effectively prevents the problem.

**Strengths:**

- The idea of Safe LoRA is simple yet effective, making it practical for mitigating safety risks in fine-tuning large language models (LLMs). The approach of achieving safety through weight space projection is both innovative and logical. This method addresses a critical issue in the fine-tuning of LLMs, where safety and alignment with human values can be compromised.
- The method’s training-free and data-free nature is a significant advantage, making it accessible and cost-effective. The simplicity of implementing Safe LoRA without the need for additional data or retraining is a notable strength.
- The methodology is well-presented, with the figure effectively conveying the core idea. The paper does a commendable job in breaking down complex concepts into understandable segments, aiding readers in grasping the significance and functioning of Safe LoRA.
- The experiment is comprehensive on different scenarios and models.

**Weaknesses:**

I didn't see major flaws in the paper.

Yet, I would like to suggest the author discuss more on the practical application scenario for Safe LoRA. For example, Safe Lora is intended to prevent unintended safety degradation in user fine-tuning or maybe can be applied in service-provider fine-tuning the model. Instead, it will not be applied for user-intended malicious fine-tuning.
Clarifying the scope of the proposed method would make the paper more sound.

**Questions:**

Do you think there would be adaptive attacks for the method? How it would be designed? Discussing some related to adaptive attacks may help make the work more comprehensive.

**Limitations:**

The authors have discussed limitations.

---

> ### Author Rebuttal · Authors · 2024-08-07
>
> Thanks for your genuine appreciation of the clarity and precious reviews on our work! We are delighted to receive a comment denoting that “The idea of $\textsf{Safe LoRA}$ is simple yet effective, making it practical for mitigating safety risks in fine-tuning LLMs.” Please see the response below as we address your comments.
>
> &nbsp;
>
> **Application Scenario**
>
> We thank the reviewer for the practical question raised! Here, we imagine two application scenarios possible for $\textsf{Safe LoRA}$: the benign users’ part and the LLM API providers’ part (such as OpenAI).
> * **Benign Users**
>
>     It is thought that benign users possess the model and can independently control the training process. Under these conditions, one use case would be companies aiming to fine-tune models based on business-related downstream data. This could be for internal employee queries or for customer service chatbots.  However, it is shown that even when benign data is used for fine-tuning, alignment may still be compromised [R1]. To prevent the model from responding to inappropriate queries from anyone, $\textsf{Safe LoRA}$ would provide help to ensure that the fine-tuned model remains aligned.
>
> * **LLM API Providers**
>
>     As an LLM API provider, users can upload their data for the LLM API provider to fine-tune, with users unable to interfere in the training process, only adjusting training parameters. In this scenario, the LLM API provider cannot spend extensive time checking whether the data is harmful but also wants to avoid the model from generating inappropriate responses after fine-tuning the user's data. Therefore, the LLM API provider would need to use $\textsf{Safe LoRA}$ to ensure that the model can withstand problematic queries while preserving the utility of the user's fine-tuning data.
>
> &nbsp;
>
> [R1] Xiangyu Qi, Yi Zeng, Tinghao Xie, Pin-Yu Chen, Ruoxi Jia, Prateek Mittal, and Peter Henderson. Fine-tuning aligned language models compromises safety, even when users do not intend to! In The Twelfth International Conference on Learning Representations (ICLR 2024).
>
> &nbsp;
>
> **Adaptive Attacks**
>
> We thank the reviewer for such a question and also deem the attack discussion necessary.
>
> Based on the previous application scenarios, adaptive attacks might be most plausible in the LLM API provider context. In the LLM API provider scenario, attackers are permitted to only upload malicious data and adjust training hyper-parameters (if allowed) but don’t have access to information about the model (such as weights or architecture) or interfere with the intermediate training process. We assume that attackers are aware that the API providers will use $\textsf{Safe LoRA}$ for realignment but do not know the exact alignment matrix.
>
> We propose two possible adaptive attack methods as the following:
>
>
> * **Method I - Knowledge Transfer Attack**
>
>      Without any knowledge of the private model. The attacker can only obtain the alignment vector on the open-source model. Here, the goal is to create malicious data such that the cosine similarity between the gradient during training and the alignment vector is high, while still keeping the training loss low. This approach would ensure a decrease in the model's safety. The attacker then provides the LLM API provider with crafted malicious data, i.e. similar to a transfer attack designed to bypass $\textsf{Safe LoRA}$.
>
>      However, to craft such data, the attackers should find a noise such that when applied to the text embedding (since LLMs convert discrete prompts into continuous embeddings) would result in the increase of cosine similarity that eventually evades the $\textsf{Safe LoRA}$ projection. Once the noise is identified, the attacker still needs additional steps of discrete-continuous optimizations to select the optimal discrete prompts for the generated noise, which can be done using genetic algorithms (GA) or other methods.
>
>     Nevertheless, this attack method is time-consuming as it requires optimizing the noise for each iteration of the training process. Furthermore, converting malicious embeddings into prompts using GA also incurs significant time costs. Lastly, the surrogate alignment matrix might be very different from the target API, causing attacks to fail.
>
>
> * **Method II - Model Inversion Attack**
>
>      Once again, the attacker can obtain the alignment vector by computing it on an open-source model. Since the dimension of this alignment vector matches that of the model weights, performing model inversion on the alignment vector might be able to generate data that represents safety. By mixing malicious data with the safety data obtained from model inversion and then uploading to the API provider, it might also be possible to bypass $\textsf{Safe LoRA}$. However, it is not known whether the alignment vector could be meaningful in model inversion or are current model inversion techniques on LLMs [R2] strong enough for such adaptive attacks.
>
> &nbsp;
>
> [R2] Morris, John X., et al. "Language model inversion." (In ICLR 2024)

---

> > ### Comment · Reviewer_d4Du · 2024-08-09
> > **Thanks for the rebuttal**
> >
> > Thanks for the rebuttal, which answers my questions. I increase the score to 7.

---

> > > ### Author Response · Authors · 2024-08-09
> > > **Thanks for the Feedback**
> > >
> > > We are grateful for the reviewer's prompt response and for the decision to increase the score. The use cases for Safe LoRA and the potential adaptive attacks are indeed crucial aspects. We will incorporate these discussions into our paper to make it more comprehensive.

---

### Official Review · Reviewer_93t8 · 2024-07-10

**Soundness:** 2
**Presentation:** 2
**Contribution:** 2
**Rating:** 5
**Confidence:** 4

**Summary:**

The paper proposed a post-hoc fine-tuning projection method which utilizes the aligned and unaligned weights of the model to compute the projection matrix. The method is simple (one-liner patch) and training-free.  Extensive experiments showed the effectiveness of the proposed method.

**Strengths:**

1. The paper is well-written and easy to follow
2. The motivation is sound which tries to address the problem of decreased safety after fine-tuning

**Weaknesses:**

1. The evaluation is limited. It only focuses on the Llama family which shares very similar architecture

2. The proposed method is named Safe LoRA without justifying that the projection matrix is indeed related to safety. It seems like the main goal of the paper is to constrain the fine-tuned weights to be within a limit of the original weights.

3. The applicability of the method is unknown. The method seems to highly depend on the fine-tune dataset, e.g. if users want to fine-tune the model to become safer (with all benign fine-tuning datasets) since not all models are equally safety-aligned [1] from the beginning, the proposed method will actually hinder its safety.


[1] Xie, Tinghao, et al. "SORRY-Bench: Systematically Evaluating Large Language Model Safety Refusal Behaviors." arXiv preprint arXiv:2406.14598 (2024).

**Questions:**

1. in table 4, why the utility, harmfulness and ASR score become worse (even for LoRA, especially the utility score) compared to baseline reported in table 2? If fine-tuning on this dataset makes things worse, should we consider evaluating on some other datasets?

2. per my comments in the weakness, although the work is a one-liner patch, the applicability of it is unknown.

3. in both table 4 and table 5, the authors reported decreased harmfulness score and ASR and increased MT-Bench and utility score, does this imply that only harmful data will cause strong (thresholded by $\tau$) dis-similarity through the fine-tuned process?

4. what's the result on other models such as Mistral, Phi and Gemma?

**Limitations:**

The author addressed several limitations of the method. However, I think there are more limitations as mentioned in the weakness and questions section.

---

> ### Author Rebuttal · Authors · 2024-08-07
>
> We thank the reviewer for recognizing the work and stating that “The motivation is sound which tries to address the problem of decreased safety after fine-tuning”. We will address and justify some concerns raised by the reviewer in the comment below.
>
> &nbsp;
>
> **Insights for $\textsf{Safe LoRA}$**
>
> Regarding the safety representation in the alignment matrix, we note from other sources such as [R1, R2] that considered exploring the safety landscape and task arithmetics which are a recent trend that focuses on the interpretability of weight semantics.
>
> Following the pipeline, the projection matrix is constructed by treating weight space as a normed vector space and extracting the “alignment” semantics from the difference between the aligned and unaligned models. **By constructing the alignment matrix, we essentially create a hyperspace of alignment and by selective projection, we are able to preserve both the utility and safety.** Therefore, the projection of the trained LoRA weights is intended to map them into the alignment hyperspace rather than merely constraining the fine-tuned weights to remain within a limit of the original weights.
>
> &nbsp;
>
> [R1] Wei, Boyi, et al. "Assessing the brittleness of safety alignment via pruning and low-rank modifications." (In ICML 2024).
>
> [R2] Ilharco, Gabriel, et al. "Editing models with task arithmetic." (In ICLR 2023)
>
> &nbsp;
>
> **Performance and Settings**
>
> * **Regarding Table 2 & 4**
>
>     The decrease in utility observed is primarily due to the impact of fine-tuning on the generalization ability [R3]. MT-Bench evaluates the overall performance of LLMs, and fine-tuning can negatively affect this broader capability. Consequently, the results in Table 4 have shown a decline. Additionally, the large difference in the number of training samples—520 times greater in Table 4 compared to Table 2—leads to models that are more specifically adapted to the training data, further contributing to the observed decrease in utility.
>
> * **Regarding Table 4 & 5**
>
>     On the other hand, we note that the reviewer holds the correct sense. That is, the more harmful data exist, the more layers we may need to project to maintain the alignment.
>
> &nbsp;
>
> [R3]: Yang, Haoran, et al. "Unveiling the Generalization Power of Fine-Tuned Large Language Models." (In NAACL 2024)
>
> &nbsp;
>
> **Application Scenario**
>
> We thank the reviewer for the question. Due to word limits, please see the general response for the application scenario of $\textsf{Safe LoRA}$.
>
> &nbsp;
>
> **Safe Fine-tuning**
>
> To address the reviewer's concerns on safe fine-tuning, we conduct experiments based on the reviewer's examples to demonstrate that $\textsf{Safe LoRA}$ does not heavily depend on the fine-tuned dataset.
>
> We use the Mistral model, which initially has poor alignment with a harmfulness score of 2.003, as shown in the following Table. We use the safety instruction dataset [R4] to fine-tune the model and improve its safety. As a result, the harmfulness score decreased to 1.003. Meanwhile, when applying $\textsf{Safe LoRA}$ to the fine-tuned model with $\tau = 0.5$ (projecting 5 layers), the harmfulness score is 1.012. Compared to the original model, applying $\textsf{Safe LoRA}$ still leads to an improvement in alignment.
>
>
> |   Fine-tuned Method   | Harmfulness Score |
> |:---------------------:|:-----------------:|
> | None (original model) |       2.003       |
> |         LoRA          |       1.003       |
> |       $\textsf{Safe LoRA}$       |       1.012        |
>
> &nbsp;
>
> [R4] Federico Bianchi, Mirac Suzgun, Giuseppe Attanasio, Paul Rottger, Dan Jurafsky, Tatsunori Hashimoto, and James Zou. Safety-tuned LLaMAs: Lessons from improving the safety of large language models that follow instructions. (In ICLR 2024).

---

> ### Comment · Reviewer_93t8 · 2024-08-13
> **thanks authors**
>
> thank the authors for the rebuttal. After reading it and other reviewer's comments, I have raise my score to 5. My concerns are mostly resolved. I didn't give a higher score because I still feel a deeper understanding of the method should be pursued and the application scenario is not fully convincing. I didn't give a lower score because as an academic paper, I feel there is merit in accepting this paper. good luck!

---

> ### Author Response · Authors · 2024-08-13
> **Thanks for the Feeback and More Explanation**
>
> We appreciate the reviewer’s feedback, and we are glad to know that we have addressed most of your concerns. Although there are still some concerns about the $\textsf{Safe LoRA}$ method itself and its related application scenarios, we will provide the following explanation in hopes of helping you gain a better understanding.
>
> &nbsp;
>
> **Deeper Insights of $\textsf{Safe LoRA}$**
>
> Currently, there are many related papers dedicated to manipulating models with arithmetic operations to enhance performance or add new functionalities to the models. We can use this concept to explain the working mechanism of $\textsf{Safe LoRA}$, where we improve model safety by projecting unaligned weights onto an aligned hyperspace. In $\textsf{Safe LoRA}$, aligned vectors play a crucial role. If one obtains a vector $A$ by subtracting the base model weights from a non-aligned model, this vector $A$ itself does not represent alignment. Using such a vector to create a projection matrix will not enhance safety when projecting fine-tuned weights onto it. This is why we need to derive the so-called aligned vectors from an aligned model. We do not arbitrarily manipulate the fine-tuned weights, nor do we simply restrict the distance of the fine-tuned weights from the original ones. Restricting this distance alone can lead to a decrease in utility and does not guarantee an improvement in safety. In summary, $\textsf{Safe LoRA}$ enhances safety by manipulating fine-tuned weights that are **further** from the aligned direction, pulling them back into the aligned hyperspace while maintaining utility.
>
> &nbsp;
>
> **Application Scenario**
>
> We will provide a more detailed explanation of the two application scenarios we mentioned earlier.
>
> * **Benign User**
>
>    First, we explain why a benign user would need to use $\textsf{Safe LoRA}$. As demonstrated in the experiments from the paper, even if the training data is entirely benign, alignment can still be compromised. Therefore, $\textsf{Safe LoRA}$ is necessary to address this issue.
>
>   Next, we discuss why a benign user needs to perform fine-tuning. Although current LLMs can indeed provide very logical question-and-answer interactions, using them directly as customer service bots to answer specific questions from customers is often insufficient. This is because LLMs typically lack information relevant to a particular company. Therefore, fine-tuning an existing LLM on the user’s data becomes necessary to tailor it to the company's specific needs and information.
>
>   Finally, combining the above two points, users need to fine-tune models using their own data. However, since fine-tuning can potentially lead to a loss of alignment, and generally, users do not want their customer service bots to provide inappropriate responses—because this could damage the company's reputation and increase crime rates (as it becomes easier to access harmful information)—$\textsf{Safe LoRA}$ becomes necessary to address these concerns.
>
> * **LLM API Provider**
>
>   The most common LLM API provider is OpenAI's ChatGPT. It allows users to upload data and configure relevant training parameters to fine-tune ChatGPT. However, since users are not always well-intentioned, there is a risk that they might upload malicious data. The LLM API provider cannot individually check each piece of data for harm, as this would be too time-consuming and would negatively impact the user experience. This is different from how ChatGPT checks user inputs for appropriateness during conversations. The volume of training data can be very large, making it impractical to check each piece individually. In fact, OpenAI currently does not perform checks on users' training data. **So why is $\textsf{Safe LoRA}$ necessary?** If a user uploads malicious data that removes alignment, they can fine-tune the model and then start querying it with harmful questions, potentially obtaining inappropriate answers. This undermines the alignment efforts made by LLM API providers, as malicious users could effectively bypass these safeguards by spending a small amount of money to get an unaligned model that can provide any response they desire. This issue could potentially increase societal risks by making it easier for people to access inappropriate information.
>
> Overall, $\textsf{Safe LoRA}$ allows model owners to restore their safety guardrails in an efficient manner regardless of any harmful data present.
>
> We hope that the detailed explanations we have provided about the $\textsf{Safe LoRA}$ method and its insight, as well as its application scenarios, will enhance your understanding of $\textsf{Safe LoRA}$ and its use cases. Thank you once again for taking the time to review the explanations provided. We hope that your concerns have been addressed.

---

### Official Review · Reviewer_2xQA · 2024-07-17

**Soundness:** 3
**Presentation:** 3
**Contribution:** 3
**Rating:** 6
**Confidence:** 3

**Summary:**

This paper propose a novel training-free method, Safe LoRA, to project the original LoRA to the sadety-aligned subspace. The experimental results illustrate that the proposed method can preserve the utility of downstream task and the safety of LLM output.

**Strengths:**

Strength:

1. This paper focused on an important problem about LLM safety training.

2. The proposed method is also very easy to follow and the figure about the pipeline is very clear.

3. The authors provide some experimental results to illustrate the performance of the proposed safety Lora: preserve the utility and safety of LLM.

**Weaknesses:**

Weakness:

1. I'm still not clear why directly project the LoRA weights to safety subspace can achieve such a great performance. Maybe the authors can provide more analysis and visualization to explain it. Because the proposed method is very simple and efficient and therefore it is more important to illustrate the insights and explain the reason.

2. I think the experiment setting is special and the authors create these special settings to verify the performance (such as "we augmented the Dialog Summary dataset with 100 harmful samples." ). It would be better if the authors could evaluate the proposed method on more popular benchmarks for testing the safety issue of LLM.

**Questions:**

1. line 213: "Regarding the settings of  LoRA, we only add LoRA to the “q_proj” and “v_proj” attention layers, ". Why do we only consider the “q_proj” and “v_proj” attention layers, which is not a common setting?

---

> ### Author Rebuttal · Authors · 2024-08-07
>
> We genuinely appreciate the reviewer for the comprehensive comments concerning alignment of LLMs. We are delighted to receive the positive feedback that “The proposed method is also very easy to follow…” Please see our point-to-point response to your comments below.
>
> &nbsp;
>
> **On the explanation of $\textsf{Safe LoRA}$**
>
> We thank the reviewer for raising such an insightful question which we will try to answer below. Firstly, there is some other research [R1, R2] concerning weight semantics such as exploring the safety landscape and task arithmetics that are related to the alignment matrix that we proposed. **On the other hand, we note that in addition to the safety guardrail that was governed by the alignment matrix we also control the to-be-projected layers such that the utility can as well be preserved.** Specifically, figure 3 in the paper demonstrates the harmfulness score versus utility. $\textsf{Safe LoRA}$ maintains strong downstream task performance because we selectively set $\tau$ to control the number of projected layers rather than projecting every layer indiscriminately. This selective projection allows us to retain critical task-specific information while minimizing unnecessary alterations, resulting in better performance. In summary, our introduced alignment matrix is effective prior to guiding LoRA updates towards better safety, because it entails the effort in aligning a base model to a chat model that can refuse (some of) unsafe questions.
>
> &nbsp;
>
> [R1] Wei, Boyi, et al. "Assessing the brittleness of safety alignment via pruning and low-rank modifications." (In ICML 2024).
>
> [R2] Ilharco, Gabriel, et al. "Editing models with task arithmetic." (In ICLR 2023)
>
> &nbsp;
>
> **Clarification on the experiment setting**
>
> We believe that the reviewer has some misunderstanding with regard to this issue. In fact, adding 100 harmful samples to the Dialog Summary dataset wasn't a special setup aimed at verifying performance. **It was actually part of our broader approach to demonstrate datasets with varying levels of harmful content, ranging from purely harmful data (Pure Bad), and partially harmful (Dialog Summary) to entirely benign data (Alpaca).** This setup was not specifically tailored but rather a natural part of our experimental framework, while also adopted by many other research [R3, R4]. To demonstrate the performance of our method, we present the F1 score on the Dialog Summary dataset, which indicates that $\textsf{Safe LoRA}$ can maintain the utility of downstream tasks yet also safeguard harmful content regardless of the ratio. As for evaluating alignment, we use the benchmark proposed by [R5], which is also adopted by other works [R3]. Overall, our experimental setup wasn't specifically designed to verify performance. Instead, it was intended to demonstrate attacks of varying severity.
>
> &nbsp;
>
> [R3]: Jiongxiao Wang, Jiazhao Li, Yiquan Li, Xiangyu Qi, Muhao Chen, Junjie Hu, Yixuan Li, Bo Li, and Chaowei Xiao. Mitigating fine-tuning jailbreak attack with backdoor enhanced alignment.arXiv preprint arXiv:2402.14968, 2024.
>
> [R4]: Tiansheng Huang, Sihao Hu, Ling Liu. Vaccine: Perturbation-aware Alignment for Large Language Model. arXiv preprint arXiv:2402.01109, 2024.
>
> [R5]: Xiangyu Qi, Yi Zeng, Tinghao Xie, Pin-Yu Chen, Ruoxi Jia, Prateek Mittal, and Peter Henderson. Fine-tuning aligned language models compromises safety, even when users do not intend to! In The Twelfth International Conference on Learning Representations (ICLR 2024).
>
> &nbsp;
>
> **Clarification on the LoRA setting**
>
> We thank the reviewer for advising on the details. Here, we follow the official code of Llama [R6], where the default setting of the PEFT config uses q_proj and v_proj.
>
> &nbsp;
>
> [R6]:https://github.com/meta-llama/llama-recipes/blob/main/src/llama_recipes/configs/peft.py, Line 11

---

> > ### Comment · Reviewer_2xQA · 2024-08-12
> > **Thanks for your response**
> >
> > Thanks for your response.
> >
> > The authors have solved my concerns.

---

> > > ### Author Response · Authors · 2024-08-12
> > > **Thanks for the Feedback**
> > >
> > > We sincerely thank the reviewer for the response and we felt honored that we could solve your concern, making more understanding towards the re-alignment problem of LLMs.

---

### Author Rebuttal · Authors · 2024-08-07

We would like to first thank the reviewers for their generous advice on $\textsf{Safe LoRA}$ as it will boost our understanding on the realignment issue of LLM fine-tuning. Due to character limits, we will be answering some common concerns in the general response below.

&nbsp;

**Application Scenario (Reviewer d4Du, 93t8)**

For the application scenario, we imagine two application scenarios possible for $\textsf{Safe LoRA}$: benign users and LLM API providers (such as OpenAI).

* Benign User

    It is thought that benign users possess the model and can independently control the training process. Under these conditions, some use cases include companies aiming to fine-tune models based on business-related downstream data. This could be for internal employee queries or for customer service chatbots.  However, it is shown that even when benign data is used for fine-tuning, alignment may still be compromised [R1]. To prevent the model from responding to inappropriate queries from anyone, $\textsf{Safe LoRA}$ would provide help to ensure that the fine-tuned model remains aligned.


* LLM API Provider

    As an LLM API provider, users can upload their data for the LLM API provider to train on, with users unable to interfere in the training process, only adjusting training parameters. In this scenario, the LLM API provider cannot spend extensive time checking whether the data is harmful but also wants to avoid the model starting to generate inappropriate responses after fine-tuning on user data. Therefore, the LLM API provider would need to use $\textsf{Safe LoRA}$ to ensure that the model can withstand problematic queries while preserving the utility of the user's data.

&nbsp;

[R1] Xiangyu Qi, Yi Zeng, Tinghao Xie, Pin-Yu Chen, Ruoxi Jia, Prateek Mittal, and Peter Henderson. Fine-tuning aligned language models compromises safety, even when users do not intend to! (In ICLR 2024).

&nbsp;

**Literature Discussion (Reviewer X4Co)**

We note that several concurrent works came around the deadline which we will acknowledge and discuss in the revised version as they are all solutions towards the realignment of LLM fine-tuning. Here, we would provide a brief discussion of the previous literature mentioned by the reviewers and enclose a more detailed one in the revised version.

Firstly, [R1] focuses on solving concept forgetting which is a phenomenon observed by [R1] where other tasks’ performance would decrease when a specific one is chosen for fine-tuning. Furthermore, [R2] aims at solving the problem of malicious fine-tuning where they discover certain perturbations that can be added to search for a model that resists malicious fine-tuning. Vaccine was then introduced by [R2] to mitigate such issues. Meanwhile [R3] utilizes a safe template to prohibit malicious responses. Similarly, [R4] proposes certain conditions as guidelines for effective defenses. Lastly, [R5] focuses on the realignment of VLLMs.

&nbsp;

[R1] Mukhoti, Jishnu, et al. "Fine-tuning can cripple your foundation model; preserving features may be the solution."(In TMLR 2024).

[R2] Huang, Tiansheng, Sihao Hu, and Ling Liu. "Vaccine: Perturbation-aware alignment for large language model." (In ArXiv 2024).

[R3] Lyu, Kaifeng, et al. "Keeping llms aligned after fine-tuning: The crucial role of prompt templates." (In ArXiv 2024).

[R4] Rosati, Domenic, et al. "Immunization against harmful fine-tuning attacks." (In ArXiv 2024).

[R5] Zong, Yongshuo, et al. "Safety fine-tuning at (almost) no cost: A baseline for vision large language models." (In ICML 2024).

&nbsp;

**Performance of $\textsf{Safe LoRA}$ on Other Public Models (Reviewer 93t8)**

In response to the reviewer's comment, we performed additional experiments using the Gemma model. We conducted experiments on the Dialog Summary dataset using the same setup described in Section 4 and present the results in the Table below. **Consistent with the results from the Llama series, $\textsf{Safe LoRA}$ sacrifices little utility, with its Rouge F1 score at 46.49%, but effectively reduces the harmfulness score to 2.209.** Although SafeInstr and BEA both achieve good utility, they do not effectively improve alignment, with their harmfulness scores close to or greater than 3.

| Attack(adversarial data) | fine-tued |  Fine-tuning Method   |  Utility   | Harmfulness Score |    ASR     |
|:----------------------------:|:---------:|:---------------------:|:----------:|:-----------------:|:----------:|
|              ✘               |     ✘     | None (original model) |   32.38%   |       1.033       |   2.12%    |
|              ✘               |     ✓     |         LoRA          |   49.93%   |       1.036       |   1.52%    |
|              ✓               |     ✓     |         LoRA          |   49.95%   |       3.803       |   93.33%   |
|              ✓               |     ✓     |       SafeInstr       | **50.45%** |       3.389       |   90.61%   |
|              ✓               |     ✓     |          BEA          |   49.27%   |       2.818       |    50%     |
|              ✓               |     ✓     |       $\textsf{Safe LoRA}$        |   46.49%   |     **2.209**     | **32.42%** |

---

### Decision · Program_Chairs · 2024-09-25

**Decision:**

Accept (poster)

**Comment:**

Thanks for your submission to NeurIPS 2024.
Overall, the reviewers recognize that the paper presents a novel training-free method, Safe LoRA, which introduces the projection of LoRA weights from selected layers to the safety-aligned subspace. The experiment results demonstrate that Safe LoRA can preserve the model's safety performance across downstream tasks and mitigate the negative impact introduced by the malicious data. The reviewers agree that this paper makes important contributions to the community and is valuable to be accepted.